# Generate-then-Test: Automated Test Case Generation for WebAssembly Using Large Language Models

## Abstract

The reliability and security of WebAssembly (Wasm) binaries are crucial for modern web development, yet effective testing methodologies remain undeveloped. This paper addresses the gap in Wasm binary testing by proposing a novel approach for test cases generation, leveraging Large Language Models (LLMs) to enhance test coverage and bug detection. Traditional testing approaches typically require access to source code, which is often unavailable for Wasm binaries. Our generate-then-test methodology overcomes this limitation by generating equivalent C++ code to simulate expected Wasm behavior, creating and mutating test cases in C++, and compiling these tests to evaluate them against the Wasm binary. Key contributions include automated test case generation using LLMs and improved code coverage through type-aware mutations, with comprehensive evaluation demonstrating the effectiveness of our approach in detecting subtle bugs in Wasm binaries, thereby ensuring more reliable Wasm applications.

## 1 Introduction

WebAssembly (Wasm) is a low-level, portable bytecode language designed for high-performance computations, providing near-native execution speeds and extensive cross-platform compatibility. Initially developed for web applications (Haas et al., 2017), Wasm is now supported by all major browsers, including Chrome, Firefox, Safari, and Edge (McConnell, 2017). Its use has since expanded to various domains, including mobile devices (Pop et al., 2022), smart contracts (McCallum, 2019), blockchains (Bian et al., 2019; Protzenko et al., 2019), and the Internet of Things (Gurdeep Singh & Scholliers, 2019; Liu et al., 2021).

With its increasing adoption, the demand for in-depth analysis and testing of Wasm has become increasingly critical. While Wasm is commonly used for legitimate applications, it has also been exploited for malicious purposes, such as cryptojacking, where Wasm code is covertly executed in browsers to mine cryptocurrencies (Konoth et al., 2018; Kharraz et al., 2019; Musch et al., 2019b; Romano et al., 2020), highlighting the importance of thoroughly understanding Wasm. As a compilation target for high-level languages such as C, C++, Go, and Rust, Wasm presents unique challenges, particularly when distributed as third-party modules without access to the original high-level source code (Musch et al., 2019a; Romano & Wang, 2023). These challenges are further exacerbated by the fact that 28.8% of Wasm binaries are minified (Hilbig et al., 2021), often with obfuscated variable names and potentially flaky test suites (Liu et al., 2024b), making interpretation and testing more difficult. These issues underscore the urgent need for developing robust test suites to ensure WebAssembly modules are both benign and functionally correct.

Despite Wasm's growing significance in modern web development, effective testing methods remain underdeveloped. A key challenge in testing Wasm binaries is the lack of access to original source code and corresponding test cases, which renders traditional testing approaches that depend on source code availability not applicable (Pacheco et al., 2007; Choi et al., 2019; Han et al., 2019; Watson et al., 2020). This calls for an automated solution capable of generating relevant test cases to validate the functionality of Wasm binaries without relying on the original source code.

This paper addresses the gap in Wasm binary testing by proposing a novel approach for test case generation, leveraging Large Language Models (LLMs) to improve test coverage and bug detection.

The core of our approach lies in a generate-then-test methodology, inspired by Yu et al. (2022), designed to evaluate the functionality and correctness of WebAssembly binaries. We begin by generating functionally equivalent C++ code based on a high-level understanding of the Wasm binary's intended functionality. This step allows us to leverage the extensive toolsets and debugging capabilities available for C++, facilitating more robust and efficient testing. Moreover, using C++ as an intermediate representation streamlines test case generation and mutation, ultimately leading to a more comprehensive evaluation of the Wasm binary. This generated C++ code serves as the foundation for crafting initial test cases, covering various aspects of the binary's functionality. To improve test coverage, we apply type-aware mutations to refine these initial tests, producing diverse inputs to uncover edge cases and potential bugs. Finally, these mutated test cases are compiled with the original Wasm binary to verify its behavior across all scenarios. This approach is implemented in our tool, WasmTest, which automates test generation and validation for Wasm binaries.

In summary, our work makes the following contributions:

- **Automated Test Case Generation**: We leverage LLMs to automate the generation of intermediate C++ code and test cases from high-level descriptions of Wasm functionalities.
- **Enhanced Code Coverage**: Our type-aware mutation strategy significantly enhances test case diversity and coverage, enabling the detection of subtle bugs in Wasm binaries.
- **Evaluation and Validation**: We demonstrate the effectiveness of WasmTest through metrics such as compile rate, code coverage, test correctness, and bug detection rate.

## 2 RELATED WORKS

### 2.1 WEBASSEMBLY INTERPRETATION

In the realm of WebAssembly understanding, significant efforts have been made towards recovering data and function types, generating function summaries, and decompiling Wasm to high-level programming languages. For example, Lehmann & Pradel (2022) leveraged LSTM to recover high-level input parameters and return data types for Wasm functions. Romano & Wang (2023) developed semantics-aware intermediate representations for Wasm functions and trained effective machine learning classifiers to predict module and function purposes. Huang & Zhao (2024) pretrained a BERT language model for WebAssembly on a multi-modal corpus of C/C++ source code, natural language documentation and corresponding WebAssembly, achieving boosted performance on type recovery and code summarization. Furthermore, efforts to decompile Wasm into high-level C/C++ snippets have incorporated fine-tuning or symbolic program analysis into LLMs (She et al., 2024; Fang et al., 2024). Despite these advances, no prior work has focused on automatically generating software tests for WebAssembly, making our approach as the first to address this need.

There are several fuzzing tools that generate valid, randomized WebAssembly binaries to test runtimes or tools. Wasm-smith (Nick Fitzgerald, Alex Crichton, 2024) focuses on creating fuzzed modules for testing the robustness of runtimes, validators, or parsers; Wasm-mutate (Cabrera-Arteaga et al., 2024; Arteaga et al., 2022) mutates Wasm programs to produce semantically equivalent variants for fuzzing compilers; and Wasm-maker (Cao et al., 2024) performs differential testing of runtimes with syntactically and semantically rich binaries. These works target the infrastructure around Wasm, while our approach, WasmTest, directly generates test cases for Wasm code to verify its functionality and ensure it is bug-free.

### 2.2 LLMS FOR CODE

Recent progress in LLMs, particularly code-specific models like CodeLlama (Rozière et al., 2023), have demonstrated their remarkable proficiency in handling code-related tasks (Hou et al., 2023; Fan et al., 2023). Their ability to grasp intricate semantics and structural nuances makes them highly effective in tasks requiring deep insights into program semantics and structures. These advancements have opened new possibilities for the automated generation of test suites for code snippets. Recently, LLMs have shown strong performance in binary or assembly-level code reverse engineering (Pearce et al., 2022; Xu et al., 2023; Wong et al., 2023; Al-Kaswan et al., 2023; She et al., 2024; Fang et al., 2024). Our work builds on these advancements by applying LLMs to generate comprehensive, semantically meaningful test inputs for Wasm binaries.

Researchers have also integrated LLMs with traditional test generation techniques to enhance automated testing. CODAMOSA (Lemieux et al., 2023) uses LLMs to generate inputs when traditional fuzzing stalls; ChatUniTest (Xie et al., 2023) refines Java test cases iteratively; MuTAP (Dakhel et al., 2024) generates targeted tests with intentionally created program errors; TitanFuzz (Deng et al., 2023a) and FuzzGPT (Deng et al., 2023b) specialize in creating inputs for deep learning libraries; and Fuzz4All (Xia et al., 2024) provides a generic framework for generating tests across programming languages using documentation or specifications.

## 2.3 Automated Test Generation

Automated test generation is a widely-used approach for identifying software bugs. Black-box testing, such as fuzz testing (Miller et al., 1990; Woo et al., 2013), feeds random inputs to the system under test (SUT) or function under test (FUT) without accessing its source code (Nidhra & Dondeti, 2012; Woo et al., 2013). In contrast, white-box testing analyzes the source code to produce higher-quality test cases (Pacheco et al., 2007; Nidhra & Dondeti, 2012). For example, symbolic execution (King, 1976) overcomes coverage limitations by solving symbolic path constraints to generate tests that explore deeper paths. Coverage-guided fuzzing (Serebryany, 2016), or grey-box testing, strikes a balance by using coverage information from the SUT to refine input generation and mutation. Recently, LLMs have been used to generate semantically meaningful test inputs for problems coded in dynamically-typed languages when traditional approaches fail (Liu et al., 2024a). Our work focuses on directly generating tests for binary-level languages like WebAssembly, using LLMs to create high-quality, semantically rich test inputs (*i.e.*, white-box), which are then scaled to a larger amount through type-aware mutation (*i.e.*, black-box).

## 3 Methodology

Figure 1 provides an overview of WasmTest, which takes as input a Wasm snippet, its intended functionality summary, and optionally example tests. WasmTest begins by constructing a prompt to generate multiple functionally equivalent C++ implementations filtered by the example tests and a majority-voting approach. Then, WasmTest utilizes LLMs and the validated C++ implementations to generate seed test cases. To enhance test coverage, type-aware mutations are applied to the seed tests, modifying input values while preserving their types to explore edge cases. Afterward, WasmTest compiles the C++ test cases with the original Wasm code and evaluates them in a WebAssembly runtime. Finally, metrics are reported to evaluate the effectiveness of the generated tests.

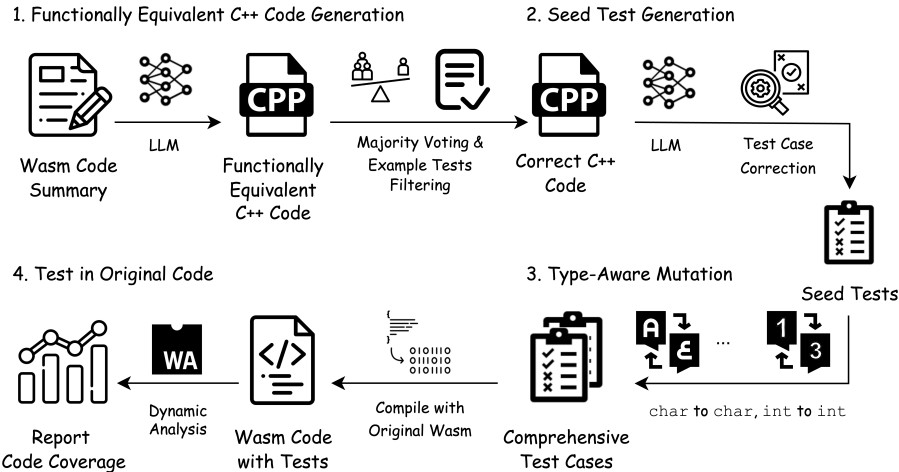

Figure 1: Overview of WasmTest.

Figure 2 is a running example of WasmTest. Wasm code summary (with example usage tests) (Figure 2a) and Wasm code under test (Figure 2b) are the inputs to the pipeline. This Wasm code is intended to remove elements from the first list that are present in the second list. Based on the summary, we first prompt the LLM to generate a few functionally equivalent C++ code samples

```
1 / Write a function removeElements to remove
↪  all elements from a given list present
↪  in another list.
2 /**
3 * In: {1, 2, 3, 4, 5, 6, 7}, {2, 4, 6}
4 * Out: {1, 3, 5, 7}
5 */
```

(a) Wasm code summary.

```
1 (func $removeElements (type 2) (param i32
↪  i32 i32)
2   (local i32 i32 i32 i32 i32 i32)
3   local.get 0
4   i32.const 0
5   i32.store offset=8
6   (;...;)
7   loop  ;; label = @1
8     (;...;)
9   end)
```

(b) Wasm code under test.

```
1 vector<int> removeElements(vector<int> list1,
↪  vector<int> list2) {
2   vector<int> result;
3   for(int i=0; i<list1.size(); i++){
4     if(find(list2.begin(), list2.end(),
↪      list1[i]) == list2.end())
5       result.push_back(list1[i]);
6   }
7   return result;
8 }
```

(c) Function code generated by Codeqwen1.5-7B.

```
1 vector<int> l1 = {1, 2, 3, 4, 5, 6, 7, 8, 9,
↪  10};
2 vector<int> l2 = {5, 7};
3 vector<int> new_3 = {1, 2, 3, 4, 6, 8, 9, 10};
4 assert((removeElements(l1, l2)  == new_3));
5 // Mutated tests
6 vector<int> new_5 = {1, -8, 3, 4, 6, 8, 9, 10};
7 assert((removeElements(vector<int> {1, -8, 3,
↪  4, 5, 6, 7, 8, 9, 10}, vector<int> {5, 5,
↪  7})  == new_5));
```

(d) Generated tests after mutation and correction.

Figure 2: A running example of WasmTest.

(Figure 2c). Next, based on the correct implementations, we prompt the LLMs to generate high-quality seed test cases (line 1-4 on Figure 2d). These seed tests are then mutated according to their data types (line 5-7, highlighted on Figure 2d) and corrected to ensure proper output. Finally, we compile the tests in C++ with the original Wasm code (Figure 2b) and run dynamic analysis to collect test coverage and bug detection statistics.

### 3.1 GENERATING FUNCTIONALLY EQUIVALENT C++ CODE.

The first step in WasmTest is generating C++ code that accurately represents the Wasm binary's high-level functionality. This step is critical as it enables the use of C++'s extensive toolsets and debugging capabilities, thereby enhancing the robustness of correctness validation. The input to this process is a natural language description of the Wasm snippet's functionality, which can be provided directly or derived through summarization and decompilation techniques (Fang et al., 2024; She et al., 2024; Huang & Zhao, 2024). WasmTest constructs a code generation prompt by combining few-shot demonstration examples with the functionality summary and feeds it into a LLM. The LLM generates multiple C++ implementations, which are then validated using example tests. Additionally, a majority-voting filtering step is employed, where an implementation is considered correct only if it produces the same output as more than half of the other generated implementations for the given use cases. Only those implementations deemed correct proceed to the next phase.

### 3.2 GENERATING SEED C++ TEST CASES

After generating the correct C++ implementations, WasmTest proceeds to create seed test cases. We restrict the LLMs to generating test inputs only, excluding their suggested expected outputs due to observed inconsistencies in their output values. To address this limitation, WasmTest runs these LLM-generated inputs through the validated C++ implementations (as discussed in Section 3.1) to compute the correct expected outputs. This guarantees that the tests faithfully reflect the code's actual behavior. By taking this approach, WasmTest capitalizes on the LLM's strengths in generating high-quality test inputs, while mitigating potential inaccuracies in its output generation.

### 3.3 TYPE-AWARE MUTATION OF C++ TEST CASES

To further enhance test coverage, WasmTest applies type-aware mutations to the seed test cases, generating a wider array of inputs that can reveal edge cases and potential bugs. This type-aware mutation strategy, guided by Liu et al. (2024a), systematically alters test inputs based on the data types involved in the Wasm snippet. The mutation rules, listed in Table 1, are governed by the types and structures defined in the code summary and the C++ implementations. For example, if the

Table 1: List of basic type-aware mutations over input $x$.

| Type | Mutation | Type | Mutation |
|------|----------|------|----------|
| `int/float` | Returns $x \pm randint(-10, 10)$ | `List` | Remove/repeat a random item $x[i]$
Insert/replace $x[i]$ with `Mutate`($x[i]$) |
| `bool` | Returns a random boolean | `Tuple` | Returns `Tuple(Mutate(List(x)))` |
| `NoneType` | Returns `None` | `Set` | Returns `Set(Mutate(List(x)))` |
| `str` | Remove a sub-string $s$
Repeat a sub-string $s$
Replace $s$ with `Mutate`($s$) | `Dict` | Remove a key-value pair $k \rightarrow v$
Update $k \rightarrow v$ to $k \rightarrow$ `Mutate`($v$)
Insert `Mutate`($k$) $\rightarrow$ `Mutate`($v$) |

Table 2: Data statistics for HumanEval-X and MBXP benchmarks from Zan et al. (2023).

| Benchmark | Num. | Working Num. | S. PL | T/N. | W/C. | W/L. | S/C. | S/L. |
|-----------|------|--------------|-------|------|------|------|------|------|
| HumanEval-X (2023) | 164 | 161 | C++ | 7.8 | 6784.9 | 295.2 | 252.5 | 10.4 |
| MBXP (2022) | 974 | 773 | C++ | 3.1 | 6063.9 | 160.1 | 192.9 | 9.2 |

input is a string, mutations may introduce variations such as altering string lengths, adding special characters, or changing case sensitivity. Similar to Section 3.2, these mutated test cases are passed to the validated C++ implementations to ensure the validity of the tests. These mutations ensure that the generated test cases probe deeper into the logic of the Wasm code, uncovering flaws missed by initial seed tests. As LLM-generated tests may come in different forms, our mutation can handle the all types of assertion tests observed across models, including direct function call (`isPrime(n) == true`), container elements test (`v[0] == 10`), variable comparison (`result == expected`), and assertions joined by the logical AND (`a == 1 && b == 2`).

### 3.4 COMPILING C++ TESTS TO WASM FOR TEST EXECUTION

With a comprehensive set of C++ tests ready, WasmTest compiles these tests using Emscripten, enabling test execution against the original Wasm snippet in a WebAssembly runtime. To gauge the test suite's thoroughness, WasmTest reports C++ and WebAssembly code coverage, with high coverage across both languages indicating thorough testing of the code's functionality.

## 4 EXPERIMENTS

### 4.1 EXPERIMENTAL SETUP

When selecting evaluation datasets, two key factors guide our decision. First, the datasets must be compilable to WebAssembly. Second, they must include natural language descriptions. To meet these criteria, we utilize two code benchmark datasets. Each dataset sample provides example usage, which we convert into example tests, and a corresponding natural language task description, making them well-suited for our evaluation.

**HumanEval-X** HumanEval-X (Zheng et al., 2023) is a widely adopted Natural Language to Code (NL2Code) dataset which has a split in C++. It is an adaptation of the HumanEval (Chen et al., 2021) dataset, which is curated by human experts, therefore reducing the risk of data contamination.

**MBXP** MBXP (Athiwaratkun et al., 2023) is another NL2Code dataset that includes a C++ split. It is generated using a parsing-based conversion framework that converts Python snippets in MBPP (Austin et al., 2021) to C++.

Detailed dataset statistics are presented in Table 2, which outlines key metrics including the number of instances (*Num.*), the number of instances that successfully compile to WebAssembly (*Working Num.*), the solution's programming language (*S.PL*), the average number of test cases per function (*T.N.*), and metrics that capture the average number of characters and lines for compiled WebAssembly Text (WAT) files (*W.C.*, *W.L.*), as well as canonical solutions (*S.C.*, *S.L.*).

The models that we focus on are listed in Table 3. We include 7 open-source LLMs. The models include both general-purpose LLMs and code-specific LLMs, with sizes ranging from 3 to 16 billion parameters. We evaluate WasmTest by integrating 7 LLMs into the testing pipeline, utilizing default settings with 0.8 temperature and generating three samples for each task. We implement LLM inference with vLLM (Kwon et al., 2023) and PyTorch 2.2.1 (cuda 12.1). We deploy them onto a server with an AMD EPYC Milan 7643 48-Core CPU@2.30GHz, 1TB RAM, and an NVIDIA L40 ADA 48GB GPU.

Table 3: Large language models used.

| Model | Parameter Size | Context Length |
|---|---|---|
| DeepSeek-Coder-V2-Lite-Instruct (Zhu et al., 2024) | 16B | 128k |
| DeepSeek-Coder-Instruct-v1.5 (Guo et al., 2024) | 7B | 4k |
| Starcoder2 (Lozhkov et al., 2024) | 3B,7B | 16k |
| CodeQwen1.5-Chat (Bai et al., 2023) | 7B | 64k |
| Qwen1.5-Chat (Bai et al., 2023) | 14B | 32k |
| Meta-Llama-3-Instruct (AI@Meta, 2024) | 8B | 8k |

## 4.2 EVALUATION METRICS

**Compile Rate.** We measure the percentage of generated test cases that successfully compile with both the original C++ and WebAssembly code under test. For C++ compilation, we use GCC with the `-std=c++17` flag to compile the tests alongside the function code. For WebAssembly, we use Emscripten with the `-sMAIN_MODULE=1` flag, compiling the test cases as the main module, aligning function name, and linking them to the WebAssembly code. We report the percentages of successfully compiled test cases in C++ (% C++ Compile) and WebAssembly (% Wasm Compile).

**Code Coverage.** Code coverage, also referred to as test coverage, is a metric that measures how much of a program's source code is executed during testing. Higher code coverage generally suggests a lower likelihood of undetected software bugs. We measure two types of coverage: C++ code coverage and WebAssembly code coverage. For C++ code coverage, we use Gcov (Project, 2024), which can be used in conjunction with GCC to measure how frequently each line of code is executed and identify which lines are actually covered during testing. For WebAssembly, we use Wasabi (Lehmann & Pradel, 2019), a dynamic analysis tool for WebAssembly based on binary instrumentation. Wasabi inserts additional WebAssembly code between the program's original instructions to call into JavaScript-based analysis functions. We implement these analysis functions as hooks to perform operations whenever a particular instruction is executed. To obtain instruction coverage, we record the location of each executed instruction.

**Pass Rate.** Pass rate measures the percentage of generated tests that run successfully with function under test without errors. A higher pass rate indicates that the generated test cases align with the expected functionality. Our assumption is that if the Wasm binary under test is correctly implemented, all generated tests should pass, which indicate that the test inputs are valid, the expected outputs are correct, and the tests compile successfully with the original Wasm binary.

**Bug Detection Rate.** Achieving high coverage does not necessarily mean that bugs will be detected. To address this, we evaluate the effectiveness of our generated tests in detecting bugs within a subset of buggy implementations. A higher bug detection rate suggests that the generated test cases are comprehensive and robust in identifying incorrect code. In addition to evaluating existing buggy code, we also apply mutation testing to randomly selected evaluation data, manually inserting bugs into the code. Mutation testing (Budd, 1980) is considered as a more precise assessment of test quality than code coverage alone (Petrović & Ivanković, 2018). This process creates multiple artificial buggy versions of the program, called a mutant, each containing one deliberately inserted, subtle error. The test suite's quality is measured by the number of mutants it can detect, a process known as "killing" the mutants. In line with previous works (Shi et al., 2014; Liu et al., 2024a), we use the percentage of mutants killed as a key metric for evaluating bug detection rate.

Table 4: Compile and pass rates with coverage metrics.

| Model | % C++ Compile | % Wasm Compile | C++ Coverage | Wasm Coverage | Pass |
|---|---|---|---|---|---|
| | | HumanEval-X | | | |
| DeepSeek-Coder-V2 | 89.1% | 89.1% | 99.4% | 94.6% | 78.1% |
| DeepSeek-Coder-v1.5 | 85.2% | 85.2% | 99.4% | 95.2% | 80.3% |
| Starcoder2-3B | 81.7% | 81.7% | 98.7% | 94.6% | 72.9% |
| Starcoder2-7B | 91.4% | 91.4% | 99.2% | 94.8% | 85.5% |
| CodeQwen1.5-Chat | 86.4% | 86.4% | 99.3% | 95.2% | 76.5% |
| Qwen1.5-Chat-14B | 49.2% | 49.2% | 99.4% | 96.4% | 47.2% |
| Meta-Llama-3 | 88.9% | 88.9% | 99.1% | 93.4% | 81.7% |
| | | MBXP | | | |
| DeepSeek-Coder-V2 | 84.8% | 84.8% | 99.6% | 96.6% | 39.6% |
| DeepSeek-Coder-v1.5 | 81.4% | 81.4% | 89.5% | 93.5% | 45.1% |
| Starcoder2-3B | 72.2% | 72.2% | 92.2% | 96.9% | 62.3% |
| Starcoder2-7B | 80.5% | 80.5% | 93.7% | 96.6% | 61.7% |
| CodeQwen1.5-Chat | 87.6% | 87.6% | 89.6% | 91.3% | 50.0% |
| Qwen1.5-Chat-14B | 68.2% | 68.0% | 90.6% | 91.4% | 39.2% |
| Meta-Llama-3 | 90.3% | 90.3% | 88.9% | 95.5% | 57.4% |

## 5   ANALYSIS

- **RQ1 (Generation of Compilable Tests):** What is the success rate of WasmTest in generating C++ functions that pass provided example tests and in producing compilable test cases across various evaluation tasks?

- **RQ2 (Test Coverage and Correctness):** For test cases generated by WasmTest, how comprehensive is their code coverage, and do they produce accurate test outputs?

- **RQ3 (Ablation on Mutation):** How does type-aware mutation affect code coverage and compile rate metrics?

- **RQ4 (Bug Detection):** How does WasmTest improve the detection of subtle and complex bugs in WebAssembly binaries?

- **RQ5 (Accuracy of Test Case Prediction):** To what extent can a LLM accurately predict test outputs during the initial generation of test cases before any correction is applied?

### 5.1   RQ1: GENERATION OF COMPILABLE TESTS

In this research question, we evaluate WasmTest's ability to generate compilable C++ functions and test cases across evaluation tasks, using the performance of various LLMs. Figure 3 shows the valid file counts that were successfully generated and compiled at each task stage for each model.

In Figure 3, *# Tasks* represents the number of WebAssembly binaries under test. WasmTest first generates equivalent C++ function code samples (Section 3.1). The *Function Generation* phase counts tasks where at least one generated function sample passes example tests. Subsequently, LLMs generate seed test cases (Section 3.2), which undergo type-aware mutation and output correction (Section 3.3). The *Test Generation* phase indicates the number of tasks yielding a full suite of test cases. Finally, WasmTest compiles the generated test cases with the original code to keep those that are *C++ Compilable* and *Wasm Compilable*.

**Correctness of Generated C++ Function Code:** The decline in valid tasks during the *Function Generation* phase illustrates initial challenges in producing syntactically correct C++ code. As illustrated in Figure 3, model performance varies significantly. Models, such as DeepSeek-Coder and CodeQwen, achieve a compile and pass rate of approximately 79.6%, demonstrating their ability to translate functionality summaries into accurate, executable code. In contrast, some LLMs show sharp declines, indicating challenges in generating correct implementations from functionality summaries. Qwen1.5-14B showed the lowest performance on HumanEval-X, with a 36.7% compile and pass rate. StarCoder2-7B achieves just 24.6% rate on MBXP dataset.

**Compilability of Generated Test Cases:** The subsequent decline during the *Test Generation* to *C++ Compilable* transition is less steep but still notable, with an average failure rate of 18.8% across

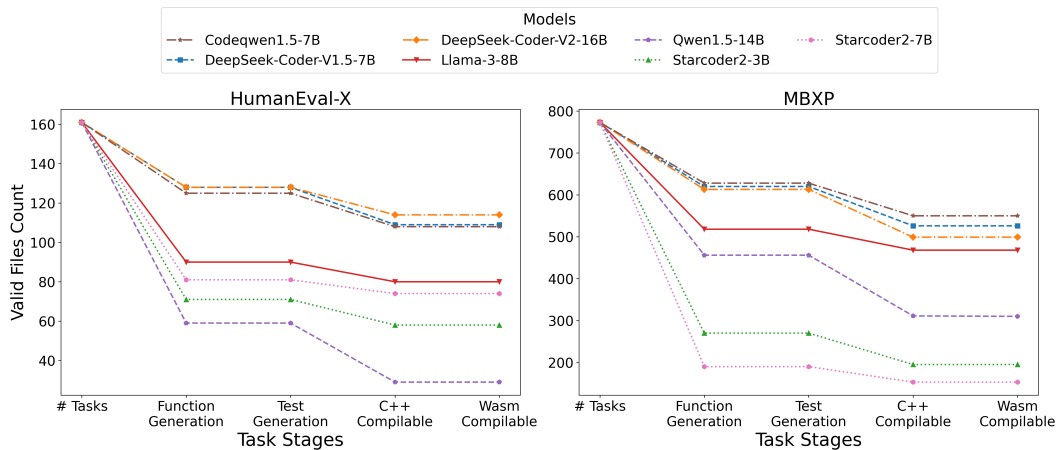

Figure 3: Number of successful, compilable generation through each stage of the WasmTest pipeline.

models. DeepSeek-Coder-V2-16B exhibits relatively better resilience in this transition, maintaining an average success rate of 87.0% across the two datasets, indicative of superior C++ syntax and semantics comprehension. In contrast, models like Qwen1.5-14B and StarCoder2-3B demonstrate significant drop-offs in this stage, where around only 77.0% and 58.7% of tasks succeed.

This analysis highlights the differing capabilities of the models in handling the complexity of generating functionally equivalent and compilable code and tests. Three models stand out in this regard: DeepSeek-Coder-v1.5-7B, DeepSeek-Coder-V2-16B, and CodeQwen1.5-7B. They consistently show fewer declines through stages compared to others, suggesting that certain models are better equipped to handle the nuances of both C++ and WebAssembly environments.

## 5.2 RQ2: TEST COVERAGE AND CORRECTNESS

This research question delves into the effectiveness of WasmTest-generated test cases, focusing on their coverage and correctness. We assess this by analyzing code coverage and pass rates from compilable test cases as detailed in Table 4. These metrics provide insights into how comprehensively the test cases exercise the code and their accuracy in validating the functionality as intended.

**Comprehensive Code Coverage:** Table 4 details the code coverage metrics for both C++ and Wasm. Overall, all models exhibit high coverage percentages, with most exceeding 90% in both environments. Notably, DeepSeek-Coder-V2 achieves nearly complete coverage on the HumanEval-X dataset, with 99.4% in C++ and 94.6% in Wasm, suggesting a robust capability to cover a wide range of code functionalities and edge cases. However, variations exist across datasets and models; for instance, Meta-Llama-3 shows slightly lower C++ coverage on the MBXP dataset (88.9%). Comparing C++ coverage with Wasm coverage, we see a high consistency across datasets and models, with the majority differ within 5%. Most code coverage metrics exceed 90%, indicating that WasmTest-generated test cases perform equally well in both C++ and Wasm environments.

**Pass Rate Analysis:** The pass rate is crucial for assessing the correctness of the generated test cases, where a high pass rate indicates that the generated tests not only compile but also execute successfully, confirming alignment with expected functional outcomes. In the HumanEval-X dataset, Starcoder2-7B and Meta-Llama-3 stand out with 85.5% and 81.7% pass rate, respectively, indicating a high level of correctness in test execution. In contrast, Qwen1.5-Chat-14B exhibits challenges, with only 47.2% of tests passing, potentially indicating issues in function implementation and inaccuracy in test output generation. The pass rates are notably lower across all models on the MBXP dataset. This is partly because some function canonical solutions are incorrect with bugs, thus leading to buggy Wasm binaries under test. This discrepancy motivates a deeper investigation into bug detection, which is discussed in Section 5.4.

Overall, the analysis reveals that WasmTest can generate highly compilable test cases with high code coverages and correctness across models. The high coverage rates across models suggest that

WasmTest is capable of generating comprehensive test cases. However, the variability in pass rates is notable. For example, Qwen1.5-Chat-14B have a significantly lower pass rate (47.2% and 39.2% respectively), suggesting that these tests may not be accurate enough for downstream tasks like code review and bug detection. This variance underscores the importance of careful model selection when generating test cases for WebAssembly binaries in achieving effective test coverage and correctness.

## 5.3 RQ3: ABLATION ON MUTATION

Table 5: Compile rate and code coverage for pre-mutated test cases for HumanEval-X benchmark.

| Model | %Wasm Compile | Wasm Coverage | %C++ Compile | C++ Coverage |
|---|---|---|---|---|
| DeepSeek-Coder-V2 | 90.6% (+1.5%) | 91.4% (-3.2%) | 93.0% (+3.9%) | 97.3% (-2.1%) |
| DeepSeek-Coder-v1.5 | 89.1% (+3.9%) | 91.0% (-4.2%) | 91.4% (+6.2%) | 94.4% (-5.0%) |
| Starcoder2-3B | 83.1% (+1.4%) | 90.6% (-4.0%) | 87.3% (+5.6%) | 93.6% (-5.1%) |
| Starcoder2-7B | 95.1% (+3.7%) | 88.9% (-5.8%) | 97.5% (+6.1%) | 95.2% (-4.0%) |
| CodeQwen1.5-Chat | 88.8% (+2.4%) | 92.6% (-2.6%) | 91.2% (+4.8%) | 97.0% (-2.3%) |
| Qwen1.5-Chat-14B | 54.2% (+5.0%) | 94.1% (-2.3%) | 55.9% (+6.7%) | 98.2% (-1.2%) |
| Meta-Llama-3 | 91.1% (+2.2%) | 90.2% (-3.2%) | 94.4% (+5.5%) | 97.9% (-1.2%) |

Table 5 presents the compile rates and code coverage metrics for pre-mutated test cases on HumanEval-X, with differences between tests before and after mutation indicated in parentheses. We observe an around 4% decrease in code coverage for pre-mutated tests across models, while the compile rate improves by approximately 5% for C++ and 3% for WebAssembly. This trade-off suggests that while mutations enhance the breadth of code coverage, they may reduce the ability of tests to compile successfully.

The compile rate drop is due to invalid input formats introduced by mutations, which sometimes create edge cases the Wasm function cannot handle. For instance, mutating a non-empty array to empty or inserting negative integers to a positive array may present challenges in the function's handling of edge inputs, leading to compilation errors. Since Wasm binaries operate at a low level, verifying whether mutated inputs are compatible with the function becomes inherently challenging. Still, the lower compile rate suggests the function could improve its handling of edge inputs.

Increased coverage indicates that mutations broaden the test scenarios, revealing hidden bugs and verifying edge cases. As our evaluation dataset is relatively simple, the coverage gains may become more significant for complex codebases. Depending on the code complexity and goals, WasmTest can adjust mutation intensity to balance compile rates and coverage.

## 5.4 RQ4: BUG DETECTION

Table 6: Bug detection rates for generated test cases.

| Model | HumanEval-X | MBXP |
|---|---|---|
| DeepSeek-Coder-V2 | 94.0% | 98.7% |
| DeepSeek-Coder-v1.5 | 93.0% | 98.3% |
| CodeQwen1.5-Chat | 93.6% | 94.4% |

To investigate the effectiveness of WasmTest in detecting bugs in WebAssembly binaries, we have identified a subset of the MBXP dataset known for containing incorrect implementations or bugs. These code snippets, often synthesized by a code generation model, pass standard example tests but frequently fail to meet the original functional objectives (Athiwaratkun et al., 2023). To further generate a rigorous testing environment, we artificially introduce bugs or logical errors into a random set of code snippets in our evaluation set. Specifically, we manually crafted 20 buggy C++ implementations for HumanEval-X and 30 for MBXP. These implementations were then compiled into WebAssembly binaries to evaluate WasmTest's bug detection capabilities. During the creation of these buggy C++ implementations, we took the perspective of novice programmers, introducing errors such as edge case failures, misinterpretations of task requirements, and hard-coded responses to potential test cases. These manually created bugs often led to deviations from the expected behavior

for specific inputs. We run the generated test cases by WasmTest to distinguish between correctly implemented and buggy code, thereby assessing its capability to identify and handle complex bug patterns. Section A.2 presents a case study that further demonstrates WasmTest's effectiveness in detecting subtle issues within these buggy implementations.

We report bug detection rates for the top three performant models in Section 5.1 in Table 6. Results for the rest models are in Appendix A.1. The results highlight high bug detection rates through the application of our generate-then-test pipeline, especially when combined with test case mutation. DeepSeek-Coder-V2 shows exemplary performance with a bug detection rate of 98.7% on the MBXP dataset, indicating a high efficacy in identifying complex bugs within WebAssembly binaries. Similarly, DeepSeek-Coder-v1.5 and CodeQwen1.5-Chat exhibit strong bug detection capabilities, though with slightly lower effectiveness compared to DeepSeek-Coder-V2. The ability of WasmTest to achieve high bug detection rates across different models sheds light on how it may improve the reliability and security of WebAssembly binaries.

## 5.5 RQ5: ACCURACY OF TEST CASE PREDICTION

Table 7: Correct test output prediction rates.

| Model | HumanEval-X | MBXP |
|---|---|---|
| DeepSeek-Coder-V2 | 40.9% | 51.0% |
| DeepSeek-Coder-v1.5 | 38.3% | 49.4% |
| Starcoder2-3B | 41.6% | 51.2% |
| Starcoder2-7B | 38.3% | 51.7% |
| CodeQwen1.5-Chat | 37.5% | 49.5% |
| Qwen1.5-Chat-14B | 37.7% | 50.4% |
| Meta-Llama-3 | 40.3% | 51.2% |

We evaluate the effectiveness of LLM-generated test cases without applying the correction process described in Section 3.2. The "correct test output prediction rates" presented in Table 7, measures how often the outputs of LLM-generated test cases align with the expected outputs based on the provided inputs without correction. We observe that the outputs generated by LLM are often inaccurate. For instance, an LLM might generate `assert(removeElements(l1, l2) == vector {1, 4, 9})` instead of the correct cases shown in Figure 2d, lines 1–4. Comparing Table 7 with the pass rates in Table 4 shows that nearly half of the passing cases have incorrect outputs, underscoring the importance of the correction process.

## 6 DISCUSSION

The generate-then-test approach leverages LLMs for automated test case generation for binary-level code like WebAssembly, significantly improving test coverage and bug detection for WebAssembly binaries. Despite these strengths, the mutation process introduces a slight reduction in compile rates, suggesting that refinement of mutation strategies is possible to maintain a balance between test thoroughness and compile success. Additionally, WasmTest currently relies on C++ as an intermediate language. When switched to other programming languages, this approach may be less applicable, especially for some low-resource languages that are not commonly seen during LLM training. Future work may aim to extend the methodology to support additional languages and evaluate its effectiveness across a wider range of real-world Wasm applications to ensure broader applicability.

## 7 CONCLUSION

This paper presents WasmTest, a novel approach for robust testing of Wasm binaries. To address the challenges of testing Wasm binaries without source code, WasmTest employs a generate-then-test methodology that uses LLMs to automatically generate equivalent C++ code, create and mutate test cases, and compile these tests for evaluation against the Wasm binary. Results show that WasmTest is capable of generating comprehensive test cases and significantly enhances the detection of subtle bugs, making it a valuable resource for developers and security analysts.

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

# A APPENDIX

## A.1 ADDITIONAL EVALUATION RESULTS

Table 8: Bug detection rates for generated test cases.

| Model | HumanEval-X | MBXP |
|---|---|---|
| DeepSeek-Coder-V2 | 94.0% | 98.7% |
| DeepSeek-Coder-v1.5 | 93.0% | 98.3% |
| Starcoder2-3B | 84.9% | 93.5% |
| Starcoder2-7B | 81.1% | 97.9% |
| CodeQwen1.5-Chat | 93.6% | 94.4% |
| Qwen1.5-Chat-14B | 100% | 100% |
| Meta-Llama-3 | 88.1% | 95.6% |

## A.2 BUG DETECTION CASE STUDY

Consider the example illustrated in Figure 2. The Wasm Code Summary describes a function intended to remove all elements from a given list that are present in another list. For the manually created buggy code in Figure 4, we implemented a C++ version that only operates correctly when the elements to be removed are in the same order as they appear in the original list. If the order differs, the function fails to remove the elements correctly. This implementation passes the example tests where both the elements to be removed and those in the given list are sorted in ascending order. In contrast, WasmTest's ability to generate and mutate tests help to detect the subtle bug. For instance, in the generated test case (see line 7 of Figure 2d), two instances of 5 appear in `list2` before 7. In the buggy implementation, the variable `j` never increments beyond 1, as there are no additional occurrences of 5 in `list1`, the removal of 7 will never happen. Thus, an Assertion Error occurs in the generated test case, highlighting WasmTest's capability to detect subtle bugs.

```
1  vector<int> removeElements(vector<int> list1, vector<int> list2) {
2      vector<int> result;
3      int i = 0;
4      int j = 0;
5      while (i < list1.size()) {
6          if (list2.size() > j) {
7              while (list1[i] == list2[j]) {
8                  i++;
9                  j++;
10                 if (list1.size() == i || list2.size() == j) {
11                     break;
12                 }
13             }
14         }
15         result.push_back(list1[i]);
16         i++;
17     }
18     return result;
19 }
```

Figure 4: Buggy code example.

