# OpenReview forum: "Generate-then-Test: Automated Test Case Generation for WebAssembly Using Large Language Models"
_ICLR.cc/2025/Conference — Submitted to ICLR 2025_

### Official Review · Reviewer_xrYm · 2024-10-18

**Soundness:** 1
**Presentation:** 3
**Contribution:** 2
**Rating:** 3
**Confidence:** 4

**Summary:**

This paper proposes WasmTest, a tool to automatically detect bugs in Wasm binaries leveraging the power of LLMs. WasmTest converts the Wasm binaries into functionally equivalent C++ code by LLMs, and then generate testing inputs based on the C++ code by combined efforts from LLMs and type-aware mutations to detect bugs in the Wasm binaries. The authors evaluated WasmTest on 2 benchmarks, the HumanEval-X and MBXP. The evaluation demonstrated that WasmTest can generate C++ code that has high pass rate; the generated test cases can achieve high code coverage and high bug detection rate in Wasm binaries on manually injected bugs.

**Strengths:**

1. The presentation of the paper is clear and easy to follow.
2. The testing inputs generation of WasmTest can generate test cases that achieve high code coverage.
3. Evaluation on multiple LLMs demonstrate the robustness against the choice of LLMs.

**Weaknesses:**

## Summarization of major weaknesses

1. The use cases of WasmTest is not clear.
2. The use of "functionally equivalent" C++ code as the testing oracle, i.e., the ground truth of testing outcomes, is dubious. WasmTest cannot guarantee the functional equivalence of the generated C++ code with the original Wasm binaries.
3. Lack of comparison with other binary-lifting tools for Wasm like wasm2c.
4. Lack of comparison with directly generating testing inputs for Wasm binaries.

## Detailed comments

**Significance**:

The use cases of WasmTest is not particularly clear. This paper targets testing the functional bugs of Wasm binaries where the original source code is not available. The paper does not clearly state the intended audiences (or users) of the proposed tool:

1. If WasmTest is intended for attackers, then targeting functional bugs may not be interesting to them, since functional bugs typically have less severe consequences and cannot be easily exploited. Also, attackers may not have access to the natural language descriptions of the Wasm binaries, which are necessary for WasmTest to work.
2. WasmTest is unlikely to be of interests to developers, as they typically have access to the original source code and can directly test the source code.

In either cases, the WasmTest is not particularly useful.

**Originality**:

The idea of using LLMs for both binary lifting and generating test cases are already explored, as mentioned in the related work section of this paper. The novelty of this paper is the combination of them into an end-to-end testing pipeline for Wasm binaries. Yet the usability of the lifting-then-testing approach is not clear. Relying on the pass rates and the majority voting to ensure the functional equivalence of the generated C++ code is not convincing. There can be cases where the generated C++ code passes all example test cases, but still not functionally equivalent to the original Wasm binaries. Without ensuring the correctness of the C++ code, employing it as the testing oracle can lead to false positives.

**Soundness**:

The evaluation of this paper requires significant improvement:

1. The authos do not mention the false positive rate of the testing outcomes. Since the generated C++ code may not be equivalent to the original Wasm binaries, a disagreement between the outcomes from the C++ code and the Wasm binaries may not necessarily indicate a bug in the Wasm binaries, but rather a functional difference.
2. The HumanEval-X and MBXP benchmarks are not complex enough. For real-world projects, the Wasm binaries can be much more complex than a few hundred lines of code. Testing on more complex Wasm binaries is necessary to demonstrate the performance of WasmTest on more realistic scenarios.
3. The evaluation does not compare generating testing inputs directly for Wasm binaries. Without such comparison, the effectiveness of lifting to C++ code is not clear.
4. The evaluation does not compare the performance of WasmTest with other binary-lifting tools for Wasm like wasm2c. Since traditional binary-lifting tools can also lift Wasm binaries to C/C++ code, the use of LLMs in WasmTest may not be necessary.
5. The authors do not evaluate the performance of WasmTest when the natural language descriptions of the Wasm binaries are not available. In cases where only the binaries are available, which can be common in practice, the performance of WasmTest is unknown.

In sum, the evaluation of this paper is not sufficient to demonstrate the effectiveness of WasmTest.

**Questions:**

1. What are the main intended audiences and use cases of WasmTest?
2. What is the false positive rate of the generated testing results?
3. How to ensure that the generated C++ code is functionally equivalent to the original Wasm binaries? Passing example tests or majority voting cannot guarantee that.
4. How does the performance of WasmTest compare with directly generating testing inputs for Wasm binaries?
5. How does WasmTest perform when the natural language descriptions of the Wasm binaries are not available?
6. What is the performance of WasmTest on more complex Wasm binaries other than the ones presented in the benchmarks?
7. How does the Wasm-to-C++ conversion compare with other binary-lifting tools for Wasm like wasm2c?

---

> ### Author Response · Authors · 2024-11-30
>
> We thank the reviewer for providing constructive feedback. Please find the detailed response below.
>
> > **Q1: What are the main intended audiences and use cases of WasmTest?**
>
> Generating tests for code snippets is crucial for ensuring software quality and reliability, yet it remains a tedious and time-consuming task for developers [1]. Similar to existing efforts in automated test generation [2, 3], our approach focuses on automatically producing a comprehensive test suite for a given program.
>
> WebAssembly (Wasm) is widely used across a wide range of applications. However, Wasm modules are often distributed as third-party binaries without accessible source code (a black-box setting) [4]. In some cases, these binaries may even be obfuscated to hide their actual functionality. According to [5], 28.8% of Wasm binaries are minified to conceal their true purpose. This poses a challenge for developers who need to integrate such Wasm modules into their applications, as they must ensure the module behaves as intended and is free of bugs. In such a setting, **developers** have no access to the source code and may require LLM-based tools for decompilation and test generation.
>
> In this black-box context, the input and output of the testing process can be described as follows:
>
> - **Input:** The program under test, accompanied by a description of its intended functionality provided in a natural language summary.
> - **Output:** A set of comprehensive unit tests, including test inputs and expected outputs, designed to validate the module’s correctness and expose potential bugs.
>
> This black-box approach is particularly valuable for **developers** working with precompiled or obfuscated Wasm binaries, enabling them to verify and trust the functionality of these components within their applications.
>
> **Reference**
>
> [1] Wang, Junjie, et al. Software testing with large language models: Survey, landscape, and vision. IEEE Transactions on Software Engineering, 2024.
>
> [2] Arghavan Moradi Dakhel, Amin Nikanjam, Vahid Majdinasab, Foutse Khomh, and Michel C Des- marais. Effective test generation using pre-trained large language models and mutation testing. Information and Software Technology, 171:107468, 2024.
>
> [3] Jiawei Liu, Chunqiu Steven Xia, Yuyao Wang, and Lingming Zhang. Is your code generated by chat- gpt really correct? rigorous evaluation of large language models for code generation. Advances in Neural Information Processing Systems, 36, 2024a.
>
> [4] Marius Musch, Christian Wressnegger, Martin Johns, and Konrad Rieck. New kid on the web: A study on the prevalence of webassembly in the wild. In Detection of Intrusions and Malware, and Vulnerability Assessment: 16th International Conference, DIMVA 2019, Gothenburg, Sweden, June 19–20, 2019, Proceedings 16, pp. 23–42. Springer, 2019a
>
> [5] Aaron Hilbig, Daniel Lehmann, and Michael Pradel. An empirical study of real-world webassembly binaries: Security, languages, use cases. In Proceedings of the web conference 2021, pp. 2696– 2708, 2021

---

> ### Author Response · Authors · 2024-11-30
>
> > **[Q2] How does the WasmTest compare with directly generating tests, existing testing methods, and other binary-lifting tools for Wasm like wasm2c?**
>
> We are grateful to the reviewer for highlighting gaps in our literature review and suggesting additional works for consideration. Based on this feedback, we have updated the related works section to include the recommended methodologies, noting that most operate under fundamentally different settings compared to our approach. Below, we address the specific concerns and provide detailed comparisons:
>
> 1. **LLM-Based Test Generation and Fuzzing Techniques**
>
> **TitanFuzz, FuzzGPT, ChatUniTest, and Fuzz4All:**
>
> TitanFuzz and FuzzGPT are specialized deep learning libraries tailored for different testing requirements and do not align with our focus on Wasm binaries. ChatUniTest focuses on Java test case generation, but some of its features are incompatible with Wasm binaries. Lastly, while Fuzz4All is a generic framework, its application is hindered by the token length of Wasm binaries and LLMs’ limited ability to comprehend raw Wasm binaries effectively.
>
> 2. **Existing Wasm Testing Techniques**
>
> Several Wasm test generation approaches, such as **WasmMaker**, **Wasm-smith**, and **Wasm-mutate**, are designed for testing Wasm runtimes and compilers. These tools primarily generate syntactically correct Wasm code to fuzz Wasm-consuming programs. In contrast, our work focuses on testing Wasm code itself: WasmTest generates unit test cases for a given Wasm code to validate if it is performing the correct functionality and is bug-free.
>
> 3. **Detailed Comparisons:**
>
> To address the reviewer’s concerns, we conducted a comparative analysis of WasmTest and related methods. The results are summarized below, showcasing WasmTest’s performance against adaptations of existing techniques:
>
> - **Wasm decompilation works comparison:**
>
> We replace *Step 1: Generating Functionally Equivalent C++ Code* with Wasm decompilation models, which convert Wasm code to equivalent C/C++ code. Specifically, we compare WasmTest with the following binary decompilation tools one at a time. For each decompiler, we use the decompiled C/C++ code as an intermediate representation and prompt the LLM to **directly generate tests** based on the decompiled version.
>
> 1. WaDec [1] + direct generation of tests
> 2. StackSight [2] + direct
> 3. Wasm2c [3] + direct
>
> - **MuTAP integration:**
>
> We incorporate MuTAP [4] into our pipeline. Following the type-aware mutation step, we manually introduce errors into the functions under test to evaluate whether existing test cases can detect these injected bugs. For errors that remain undetected, we utilize LLMs by providing the function under test, the associated test cases, and details about the undetected errors. The LLMs are then prompted to generate new test cases specifically aimed at identifying the highlighted errors.
>
> Table 1: Metrics on HumanEval-X with one of the best performing models - CodeQwen1.5-7B.
>
> | Method              | C++ Compile | Wasm Compile | C++ Coverage | Wasm Coverage | Pass Rate |
> | ------------------- | ----------- | ------------ | ------------ | ------------- | --------- |
> | WasmTest            | **86.4%**   | **86.4%**    | **99.3%**    | **95.2%**     | **76.5%** |
> | WaDec + Direct      | 27.5%       | 28.4%        | 86.2%        | 93.3%         | 24.1%     |
> | StackSight + Direct | 54.81%      | 56.3%        | 99.1%        | 93.3%         | 22.4%     |
> | Wasm2c + Direct     | 0%          | 0%           | N/A          | N/A           | N/A       |
> | MuTAP               | 77.5%       | 77.1%        | 97.8%        | 92.3%         | 35.0%     |
>
> **Key Observations:**
>
> - Wasm2c: Fails to simplify complexities in Wasm binary (verbatim transcription), making it ineffective for guiding LLMs to generate compilable test cases.
> - WaDec and StackSight: Despite being state-of-the-art LLM-based Wasm decompilers, their lower compile and pass rates indicate challenges in achieving perfect decompilation.
> - MuTAP: Achieves the best compile and pass rates by leveraging LLMs to generate equivalent C++ code based on natural language summaries, which are easier for LLMs to process.
> - WasmTest: Outperforms all baselines across metrics due to its design, which incorporates sampling, majority voting, and type-aware mutations, ensuring superior compile rates, coverage, and pass rates.
>
> **Reference**
>
> [1] She, Xinyu, Yanjie Zhao, and Haoyu Wang. WaDec: Decompiling WebAssembly Using Large Language Model.
>
> [2] Fang, Weike, et al. StackSight: Unveiling WebAssembly through Large Language Models and Neurosymbolic Chain-of-Thought Decompilation.
>
> [3] Wasm2c. https://github.com/WebAssembly/wabt/wasm2c
>
> [4] Dakhel, Arghavan Moradi, et al. Effective test generation using pre-trained large language models and mutation testing.

---

> ### Author Response · Authors · 2024-11-30
>
> > **Q3:** **How does WasmTest perform when the natural language descriptions of the Wasm binaries are not available?**
>
> In our setting, developers that aim to test a given Wasm module should have in mind the intended purpose of this module, such as computing a hash function. The test suite should be devised to evaluate whether the module (1) correctly fulfills its functionality requirements and (2) is free of bugs.
>
> When a high-level summary of the Wasm code is unavailable, reverse engineering techniques can be employed to decompile the Wasm binary into a C++ implementation. We presented our comparison with models and tools such as WaDec, StackSight, and Wasm2c in Table 1.
>
> Besides, we note that if the code summary is not of good quality, it may be refined via existing models that are capable of Wasm code summarization, such as WasmRev [1] and StackSight. Therefore, for the purposes of our evaluation, we assume that a correct and reliable summary of the Wasm code is available, as it forms the foundation for accurate test generation.
>
> [1] Huang, Hanxian, and Jishen Zhao. "Multi-modal Learning for WebAssembly Reverse Engineering." Proceedings of the 33rd ACM SIGSOFT International Symposium on Software Testing and Analysis. 2024.
>
>
>
> > **Q4: How to ensure that the generated C++ code is functionally equivalent to the original Wasm binaries?** **What is the false positive rate of the generated testing results?**
>
> We appreciate the reviewer’s insightful comment and would like to clarify the objectives and assumptions of our approach. The primary goal of our test-case generation pipeline is to verify that a WebAssembly (Wasm) binary adheres to its stated functional description, rather than performing direct code translation.
>
> To achieve this, our methodology involves generating C++ implementations from the natural language description of the intended functionality, without any reference to the Wasm binary. The task assigned to the LLM is to produce C++ code that fulfills the functional requirements outlined in the description. This approach does not rely on semantic-preserving translation; instead, it assumes that the provided code summary is accurate and reliable. Previous studies have demonstrated that LLMs perform reasonably well in similar code generation tasks.
>
> Our approach is based on the following key assumptions:
>
> - **Accuracy of the Functional Description:** The natural language description accurately reflects the intended functionality of the code.
> - **Test Case Effectiveness:** Test cases derived from the generated C++ implementations can identify discrepancies in the Wasm binary, particularly for edge cases or buggy scenarios.
>
> We acknowledge that there are **false positives** in the testing outcomes, where the Wasm binary is correct but the generated test failed. These results are presented in Table 4 (pass rate) of our draft. Specifically, we run the generated tests on the ground-truth Wasm binary compiled from canonical solutions from HumanEval-X. These canonical solutions from HumanEval-X are human-crafted and verified, so they are expected to pass all tests. The pass rate is around 80% across models, indicating a false positive rate of around 20%. These false positives are mainly caused by errors in the code generated by LLMs in Step 1 (such as failing to account for some edge cases), which propagate the error to later steps.

---

> > ### Comment · Reviewer_xrYm · 2024-12-01
> >
> > Thanks for your reply. However, I decided to keep my current rating.

---

### Official Review · Reviewer_52kk · 2024-10-20

**Soundness:** 2
**Presentation:** 3
**Contribution:** 1
**Rating:** 3
**Confidence:** 5

**Summary:**

- This paper propose a new approach, named WasmTest, that leverages LLMs to automate the generation of intermediate C++ code and test cases from high-level descriptions of Wasm functionalities.
- This paper includes a type-aware mutation strategy for mutation testing
- The evaluation results of this paper indicates the effectiveness of WasmTest through compile rate, code coverage, test correctness and bug detection rate

**Strengths:**

- This paper is clearly written and easy to comprehend
- This paper tries to address an important and difficult problem, Wasm test generation

**Weaknesses:**

Generally, I think this paper tries to address a typical software engineering task, automated test generation; however, it lacks a comprehensive empirical study about recent work, which needs to be mentioned and compared, and thus incorrectly claims that "effective testing methods remain underdeveloped" (Line 47).

Specifically, I have the following concerns:
- This paper overlook a large number of existing Wasm test generation approaches, e.g., WASMaker [1], wasm-smith [2], Wasm-mutate [3]
- The motivation of translating Wasm code into C++ code and then generate test cases is not clearly explained. Now that you utilize LLMs to comprehend the semantics of Wasm code, why not directly use LLMs to generate Wasm tests?
- This technical contribution of this paper is limited as each component in its approach has been proposed before. 1) Wasm has existing reverse engineering tool, wasm2c [5], so you need to explain why to use LLM for code translation and add comparison evaluation between your approach and wasm2c. 2) The step 3 in your approach (type-aware mutation). There exists a large portion of mutation testing approaches, even for c/c++ programs, e.g., CCmutator [4]. Additionally, the SOTA mutation testing approaches also include LLM, e.g., LEAM [6], which I suggest to consider.
- The evaluation section needs to be improved with more details about its setup. 1) Now that you leverage LLMs for code translation, you need further evaluation to show the semantic-preserving of your approach; 2) As for bug detection, you need to explain more details about your manually crafted bugs (Line 447), e.g., the fault types, the background of your involved engineers, which is important in the area of software engineering as this paper is an AI4SE one; 3).



These reference might be considered:

[1] Cao S, He N, She X, et al. WASMaker: Differential Testing of WebAssembly Runtimes via Semantic-Aware Binary Generation[C]//Proceedings of the 33rd ACM SIGSOFT International Symposium on Software Testing and Analysis. 2024: 1262-1273.

[2] Bytecode Alliance. 2023. Github wasm-tools repository. https://github.com/bytecodealliance/wasm-tools/tree/main/crates/wasm-smith

[3] Arteaga J C, Fitzgerald N, Monperrus M, et al. Wasm-mutate: Fuzzing WebAssembly compilers with e-graphs[C]//E-Graph Research, Applications, Practices, and Human-factors Symposium. 2022.

[4] Kusano M, Wang C. CCmutator: A mutation generator for concurrency constructs in multithreaded C/C++ applications[C]//2013 28th IEEE/ACM International Conference on Automated Software Engineering (ASE). IEEE, 2013: 722-725.

[5] https://github.com/WebAssembly/wabt/wasm2c

[6] Tian Z, Chen J, Zhu Q, et al. Learning to construct better mutation faults[C]//Proceedings of the 37th IEEE/ACM International Conference on Automated Software Engineering. 2022: 1-13.

**Questions:**

1. What is the difference between your WasmTest approach and existing approaches, e.g., WASMaker, wasm-smith, Wasm-mutate?
2. What is the difference between your step 1 and existing Wasm reverse engineering approaches, especially wasm2c?
3. What is the difference between your step 3 and existing mutation testing tools, e.g., CCmutator (actually mutation testing is also a matured research area in software engineering)?

---

> ### Author Response · Authors · 2024-11-30
>
> We thank the reviewer for providing constructive feedback. Please find the detailed response below.
>
> > **[Q1] More comparison with existing Wasm test generation methods and LLM based methods**
>
> We have updated the related works section to include the recommended methodologies, noting that most operate under fundamentally different settings compared to our approach.
>
> 1. **LLM-Based Test Generation and Fuzzing Techniques**
>
> **TitanFuzz, FuzzGPT, ChatUniTest, and Fuzz4All:**
>
> TitanFuzz and FuzzGPT are specialized deep learning libraries tailored for different testing requirements and do not align with our focus on Wasm binaries. ChatUniTest focuses on Java test case generation, but some of its features are incompatible with Wasm binaries. Lastly, while Fuzz4All is a generic framework, its application is hindered by the token length of Wasm binaries and LLMs’ limited ability to comprehend raw Wasm binaries effectively.
>
> 2. **Existing Wasm Testing Techniques**
>
> Existing WebAssembly (Wasm) test generation approaches mainly target **fuzzing Wasm runtimes, compilers, or validators**, rather than testing the functionality of the Wasm binary itself. Our work focuses on generating unit tests to test a given Wasm code. For example:
>
> - **WASMaker** generates complicated Wasm binaries (tests) that are both semantically rich and syntactically correct, focusing on detecting bugs or inconsistencies in Wasm runtimes.
> - **Wasm-smith** produces fuzzed Wasm modules for testing the robustness and correctness of Wasm runtimes, validators, or parsers in a black-box manner.
> - **Wasm-mutate** uses e-graphs to create fuzzed Wasm snippets to test Wasm compilers and interpreters. By mutating a given Wasm program, it generates semantically equivalent variants for Wasm compiler fuzzing.
>
> 3. **Detailed Comparisons:**
>
> We conducted a comparative analysis of WasmTest and related methods. The results are summarized below.
>
> - **Wasm decompilation works comparison:**
>
> We replace *Step 1: Generating Functionally Equivalent C++ Code* with Wasm decompilation models, which convert Wasm code to equivalent C/C++ code. Specifically, we compare WasmTest with the following binary decompilation tools one at a time. For each decompiler, we use the decompiled C/C++ code as an intermediate representation and prompt the LLM to **directly generate tests** based on the decompiled version.
>
> 1. WaDec [1] + direct generation of tests
> 2. StackSight [2] + direct generation of tests
> 3. Wasm2c [3] + direct generation of tests
>
> - **MuTAP integration:**
>
> We incorporate MuTAP [4] into our pipeline. Following the type-aware mutation step (Step 3), we manually introduce errors into the functions under test to evaluate whether existing test cases can detect these injected bugs. For errors that remain undetected, we utilize LLMs by providing the function under test, the associated test cases, and details about the undetected errors. The LLMs are then prompted to generate new test cases specifically aimed at identifying the highlighted errors.
>
> Table 1: Metrics on HumanEval-X with one of the best performing models - CodeQwen1.5-7B.
>
> | Method              | C++ Compile | Wasm Compile | C++ Coverage | Wasm Coverage | Pass Rate |
> | ------------------- | ----------- | ------------ | ------------ | ------------- | --------- |
> | WasmTest            | **86.4%**   | **86.4%**    | **99.3%**    | **95.2%**     | **76.5%** |
> | WaDec + Direct      | 27.5%       | 28.4%        | 86.2%        | 93.3%         | 24.1%     |
> | StackSight + Direct | 54.81%      | 56.3%        | 99.1%        | 93.3%         | 22.4%     |
> | Wasm2c + Direct     | 0%          | 0%           | N/A          | N/A           | N/A       |
> | MuTAP               | 77.5%       | 77.1%        | 97.8%        | 92.3%         | 35.0%     |
>
> **Key Observations:**
>
> - Wasm2c: Fails to simplify complexities in Wasm binary (verbatim transcription), making it ineffective for guiding LLMs to generate compilable test cases.
> - WaDec and StackSight: Despite being state-of-the-art LLM-based Wasm decompilers, their lower compile and pass rates indicate challenges in achieving perfect decompilation.
> - MuTAP: Achieves the best compile and pass rates by leveraging LLMs to generate equivalent C++ code based on natural language summaries, which are easier for LLMs to process.
> - WasmTest: Outperforms all baselines across metrics due to its design, which incorporates sampling, majority voting, and type-aware mutations, ensuring superior compile rates, coverage, and pass rates.
>
> **Reference**
>
> [1] She, Xinyu, Yanjie Zhao, and Haoyu Wang. WaDec: Decompiling WebAssembly Using Large Language Model.
>
> [2] Fang, Weike, et al. StackSight: Unveiling WebAssembly through Large Language Models and Neurosymbolic Chain-of-Thought Decompilation.
>
> [3] Wasm2c. https://github.com/WebAssembly/wabt/wasm2c
>
> [4] Dakhel, Arghavan Moradi, et al. Effective test generation using pre-trained large language models and mutation testing.

---

> ### Author Response · Authors · 2024-11-30
>
> > **Q2:** **Novelty in Mutation**
>
> Our mutation process builds upon prior work [1] to systematically mutate test inputs and generate diverse test cases. However, unlike previous schemes designed for human-written tests, our mutation techniques are specifically optimized for various forms of LLM-generated tests. This includes unique structures such as function calls, container elements, and logical operations.
>
> While we acknowledge the potential for more innovative mutation techniques, our current approach demonstrates strong performance, achieving approximately 95% code coverage for both C++ and Wasm. Notably, the tests are able to detect carefully crafted bugs (negative examples) almost all the time, with DeepSeek-Coder-V2 showing exemplary performance with a bug detection rate of 98.7% on the MBXP dataset.
>
> It is distinct from the techniques employed in tools like CCmutator, which operate in completely different contexts. CCmutator focuses on mutating source code at the LLVM IR level to inject concurrency-related bugs, aiming to evaluate software testing and verification tools. In contrast, our work targets test case inputs in unit test assertions. The objective of our mutation process is to systematically diversity test case inputs, enhancing the comprehensiveness of the generated test suites and enabling superior code coverage.
>
> This clear distinction highlights the unique focus of our mutation strategy, which is designed to optimize the testing process rather than simulate concurrency-related issues within the program source.
>
> [1] Jiawei Liu, Chunqiu Steven Xia, Yuyao Wang, and Lingming Zhang. Is your code generated by chat- gpt really correct? rigorous evaluation of large language models for code generation. Advances in Neural Information Processing Systems, 36, 2024a.
>
> > **Q3:**  **How does WasmTest ensure the semantic-preserving of code translation on the C++ implementation generation process?**
>
> We appreciate the reviewer’s insightful comment and would like to clarify the objectives and assumptions of our approach. The primary goal of our test-case generation pipeline is to verify that a WebAssembly (Wasm) binary adheres to its stated functional description, rather than performing direct code translation.
>
> To achieve this, our methodology involves generating C++ implementations from the natural language description of the intended functionality, without any reference to the Wasm binary. The task assigned to the LLM is to produce C++ code that fulfills the functional requirements outlined in the description. This approach does not rely on semantic-preserving translation; instead, it assumes that the provided code summary is accurate and reliable. Previous studies have demonstrated that LLMs perform reasonably well in similar code generation tasks.
>
> Our approach is based on the following key assumptions:
>
> - **Accuracy of the Functional Description:** The natural language description accurately reflects the intended functionality of the code.
> - **Test Case Effectiveness:** Test cases derived from the generated C++ implementations can identify discrepancies in the Wasm binary, particularly for edge cases or buggy scenarios.
>
> By focusing on generating valid test cases based on the stated functionality, our approach aims to evaluate whether the Wasm binary deviates from its intended behavior or contains bugs. Ultimately, the goal is to ensure that the Wasm binary complies with the functional requirements, rather than to establish a direct translation from Wasm to C++.
>
>
>
> > **Q4: More details about bug detection (fault types, background of involved engineers, etc.).**
>
> We appreciate the reviewer’s thoughtful comments. The manually crafted bugs were introduced by hard coding modifications to existing example cases in the functionality descriptions. These modifications include the following fault types:
>
> 1. **Logical Errors:** Modifications to loop conditions (e.g., changing <= to <) to introduce off-by-one errors.
> 2. **Special Case Handling:** Failure to correctly process edge inputs, such as empty strings or boundary values.
> 3. **Semantic Misinterpretations:** Incorrect implementation of functionality, such as sorting in reverse order instead of ascending.
>
> We will release these manually crafted buggy implementations alongside our pipeline. The bugs were created by authors with experience teaching entry-level programming courses at the university level. This background in teaching provided valuable insights into common mistakes made by novice programmers, which helped in designing realistic and diverse faults for this evaluation.

---

> > ### Comment · Reviewer_52kk · 2024-11-30
> >
> > Thanks for your clarification, especially the comparison between WasmTest and existing Wasm Testing Techniques. However, I decide to keep my score.

---

### Official Review · Reviewer_m8cy · 2024-10-29

**Soundness:** 2
**Presentation:** 2
**Contribution:** 2
**Rating:** 3
**Confidence:** 3

**Summary:**

The paper introduces a testing methodology for WebAssembly (Wasm) binaries that uses Large Language Models (LLMs) to overcome the absence of source code. By generating equivalent C++ code from a high-level understanding of Wasm functionalities, the methodology facilitates the creation of seed test cases via LLMs and type-aware mutation to improve coverage and detect bugs. The approach is validated through metrics like compile rate, code coverage, and bug detection rate. Two benchmarks are used in the evaluation. The paper showcases a practical application of LLMs in addressing challenges associated with Wasm binary testing.

**Strengths:**

- Originality

Wasm binary testing is an interesting and important problem. The primary innovation of the "Generate-Then-Test" methodology is its application of LLMs to generate equivalent C++ code based on abstract descriptions of Wasm functionalities.

- Quality

The proposed approach appears to work, validated using two benchmark datasets and evaluated through several standard metrics
- Clarity

The paper is overall clear and well-organised.

- Significance:

This paper offers a new way in the testing binaries distributed without source code and this might even go beyond its immediate application in testing Wasm binaries.

**Weaknesses:**

Using LLMs to generate equivalent source code from binaries constitutes the major contribution of this paper, yet it also the bottleneck of the work. It is an interesting idea but it is also a challenging problem, isn't it? This method presents difficulties in effectively translating high-level descriptions into executable test cases, and the paper does not convincingly demonstrate the efficacy of this translation.

The experimental validation is somewhat limited due to the nature of the benchmarks used. The study employs only two source code datasets that are compiled to WebAssembly before applying the proposed methodology. This approach can create an artificial scenario tailored to the solution presented in the paper, rather than addressing the more complex and varied real-world challenges found in typical WebAssembly applications.

While the methodology is shown to be effective within the constraints of the selected benchmarks, the paper does not provide a comparison with existing binary testing approaches. This raises concerns about its advantages or improvements over traditional testing techniques.

The success of the "Generate-Then-Test" methodology relies heavily on the quality and accuracy of the high-level descriptions of Wasm functionalities. Inadequate or imprecise descriptions can lead to derived C++ code and test cases that do not accurately reflect the intended functionality of the Wasm binaries. The paper should clarify how the quality of these descriptions impacts the effectiveness of the generated tests and propose strategies to ensure the reliability of these descriptions in producing meaningful test cases.

**Questions:**

- Validation of Description Quality

What methods or criteria could be used to ensure the quality and accuracy of the high-level descriptions used for generating C++ code? Can you provide detailed methodologies that validate the adequacy of these descriptions?

- Comparison with Existing Methods

How does the proposed testing approach compare with traditional binary testing methods


- Generalizability and Scalability

How would the proposed testing perform on real-world applications?


- Impact of Inadequate Descriptions

What are the observed impacts or specific examples where inadequate descriptions have led to ineffective test generation? Can the authors provide case studies or examples?

---

> ### Author Response · Authors · 2024-11-30
>
> We thank the reviewer for providing constructive feedback. Please find the detailed response below.
>
> > **[Q1] More comparison with existing testing methods**
>
> We have updated the related works section to include the some relevant methodologies, noting that most operate under fundamentally different settings compared to our approach. Below, we address the specific concerns and provide detailed comparisons:
>
> 1. **LLM-Based Test Generation and Fuzzing Techniques**
>
> **TitanFuzz, FuzzGPT, ChatUniTest, and Fuzz4All:**
>
> TitanFuzz and FuzzGPT are specialized deep learning libraries tailored for different testing requirements and do not align with our focus on Wasm binaries. ChatUniTest focuses on Java test case generation, but some of its features are incompatible with Wasm binaries. Lastly, while Fuzz4All is a generic framework, its application is hindered by the token length of Wasm binaries and LLMs’ limited ability to comprehend raw Wasm binaries effectively.
>
> 2. **Existing Wasm Testing Techniques**
>
> Several Wasm test generation approaches, such as **WasmMaker**, **Wasm-smith**, and **Wasm-mutate**, are designed for testing Wasm runtimes and compilers. These tools primarily generate syntactically correct Wasm code to fuzz Wasm-consuming programs. In contrast, our work focuses on testing Wasm code itself: WasmTest generates unit test cases for a given Wasm code to validate if it is performing the correct functionality and is bug-free.
>
> 3. **Detailed Comparisons:**
>
> To address the reviewer’s concerns, we conducted a comparative analysis of WasmTest and related methods. The results are summarized below, showcasing WasmTest’s performance against adaptations of existing techniques:
>
> - **Wasm decompilation works comparison:**
>
> We replace *Step 1: Generating Functionally Equivalent C++ Code* with Wasm decompilation models, which convert Wasm code to equivalent C/C++ code. Specifically, we compare WasmTest with the following binary decompilation tools one at a time. For each decompiler, we use the decompiled C/C++ code as an intermediate representation and prompt the LLM to generate tests based on the decompiled version.
>
> 1. WaDec [1] + direct generation of tests
> 2. StackSight [2] + direct generation of tests
> 3. Wasm2c [3] + direct generation of tests
>
> - **MuTAP integration:**
>
> We incorporate MuTAP [4] into our pipeline. Following the type-aware mutation step (Step 3), we manually introduce errors into the functions under test to evaluate whether existing test cases can detect these injected bugs. For errors that remain undetected, we utilize LLMs by providing the function under test, the associated test cases, and details about the undetected errors. The LLMs are then prompted to generate new test cases specifically aimed at identifying the highlighted errors.
>
> Table 1: Metrics on HumanEval-X with one of the best performing models - CodeQwen1.5-7B.
>
> | Method              | C++ Compile | Wasm Compile | C++ Coverage | Wasm Coverage | Pass Rate |
> | ------------------- | ----------- | ------------ | ------------ | ------------- | --------- |
> | WasmTest            | **86.4%**   | **86.4%**    | **99.3%**    | **95.2%**     | **76.5%** |
> | WaDec + Direct      | 27.5%       | 28.4%        | 86.2%        | 93.3%         | 24.1%     |
> | StackSight + Direct | 54.81%      | 56.3%        | 99.1%        | 93.3%         | 22.4%     |
> | Wasm2c + Direct     | 0%          | 0%           | N/A          | N/A           | N/A       |
> | MuTAP               | 77.5%       | 77.1%        | 97.8%        | 92.3%         | 35.0%     |
>
> **Key Observations:**
>
> - Wasm2c: Fails to simplify complexities in Wasm binary (verbatim transcription), making it ineffective for guiding LLMs to generate compilable test cases.
> - WaDec and StackSight: Despite being state-of-the-art LLM-based Wasm decompilers, their lower compile and pass rates indicate challenges in achieving perfect decompilation.
> - MuTAP: Achieves the best compile and pass rates by leveraging LLMs to generate equivalent C++ code based on natural language summaries, which are easier for LLMs to process.
> - WasmTest: Outperforms all baselines across metrics due to its design, which incorporates sampling, majority voting, and type-aware mutations, ensuring superior compile rates, coverage, and pass rates.
>
>
>
> **Reference**
>
> [1] She, Xinyu, Yanjie Zhao, and Haoyu Wang. "WaDec: Decompiling WebAssembly Using Large Language Model." Proceedings of the 39th IEEE/ACM International Conference on Automated Software Engineering. 2024.
>
> [2] Fang, Weike, et al. "StackSight: Unveiling WebAssembly through Large Language Models and Neurosymbolic Chain-of-Thought Decompilation." Forty-first International Conference on Machine Learning.
>
> [3] Wasm2c. https://github.com/WebAssembly/wabt/wasm2c
>
> [4] Dakhel, Arghavan Moradi, et al. Effective test generation using pre-trained large language models and mutation testing.

---

> ### Author Response · Authors · 2024-11-30
>
> > **Q2:**  **Impact of inadequate or absent High-level Code Descriptions**
>
> In our setting, developers that aim to test a given Wasm module should have in mind the intended purpose of this module, such as computing a hash function. The test suite should be devised to evaluate whether the module (1) correctly fulfills its functionality requirements and (2) is free of bugs.
>
> When a high-level summary of the Wasm code is unavailable, reverse engineering techniques can be employed to decompile the Wasm binary into a C++ implementation. We presented our comparison with models and tools such as WaDec, StackSight, and Wasm2c in Table 1.
>
> Besides, we note that if the code summary is not of good quality, it may be refined via existing models that are capable of Wasm code summarization, such as WasmRev [1] and StackSight. Therefore, for the purposes of our evaluation, we assume that a correct and reliable summary of the Wasm code is available, as it forms the foundation for accurate test generation.
>
> [1] Huang, Hanxian, and Jishen Zhao. "Multi-modal Learning for WebAssembly Reverse Engineering." Proceedings of the 33rd ACM SIGSOFT International Symposium on Software Testing and Analysis. 2024.

---

### Official Review · Reviewer_2aEW · 2024-11-03

**Soundness:** 1
**Presentation:** 2
**Contribution:** 2
**Rating:** 3
**Confidence:** 4

**Summary:**

This paper introduces WasmTest, an approach for testing of WebAssembly (Wasm) binaries. To overcome the challenges of testing Wasm binaries without source code, WasmTest leverages a generate-then-test methodology. It utilizes large language models (LLMs) to automatically generate equivalent C++ code, create and mutate test cases, and compile these tests for execution against the Wasm binary. Results demonstrate that WasmTest  generates comprehensive test suites.

**Strengths:**

1. Novel Approach: Introduces a novel generate-then-test approach for black-box testing of WebAssembly (Wasm) binaries, leveraging the power of large language models (LLMs).
2. Effective Coverage: Demonstrates high test coverage for simple, single-program benchmarks.
3. Ease of Understanding: The proposed technique is straightforward and easy to comprehend.

**Weaknesses:**

- **Limited Literature Review:** The paper lacks a comprehensive review of related work, particularly in the areas of LLM-based automated test generation (e.g., CODAMOSA, ChatUniTest, MuTAP) and LLM-based fuzzing (e.g., TitanFuzz, FuzzGPT, Fuzz4All). Additionally, the review of existing Wasm testing techniques is insufficient.
- **Inadequate Test Oracle:** The reliance on LLM-generated oracles, which exhibit approximately 50% accuracy, limits the practical applicability of the approach. The black-box testing paradigm may not be suitable for developer-focused scenarios, and general fuzzing techniques with sanitizers might be more effective for attacker-focused use cases.
- **Insufficient Evaluation of Generate-Then-Test:** The paper lacks a thorough evaluation of the generate-then-test approach. It's unclear whether directly generating tests for Wasm binary code using LLMs or applying simple fuzzing techniques like AFL or libfuzzer would yield comparable or superior coverage results. Additionally, it lacks an ablation study exploring the substitution of the first component with a simpler reverse engineering approach.
- **Lack of Novelty in Mutation:** The mutation techniques employed in the paper do not appear to be significantly novel.
- **Limited Evaluation Scope:** The evaluation is restricted to a single-program dataset, which may not accurately reflect real-world application scenarios

**Questions:**

- What is the result of directly generating tests for Wasm binaries?
- Could you elaborate on the applications of this black-box setting? Specifically, who is expected to test the Wasm binaries in such a context?
- Could you provide more details on how equivalence with the C++ implementation is verified? Additionally, what if we apply reverse engineering to the Wasm binary and use the (likely non-equivalent) results to guide test case generation?
- How can the generated tests be used to detect bugs? The current 50% correct rate of the oracles generated by LLMs can definitely not be applied.

---

> ### Author Response · Authors · 2024-11-30
>
> We thank the reviewer for providing constructive feedback. Please find the detailed response below.
>
> > **[Q1] Additional comparisons with Wasm reverse-engineering tools, LLM-based test case generation models, and fuzzing techniques**
>
> We are grateful to the reviewer for highlighting gaps in our literature review and suggesting additional works for consideration. Based on this feedback, we have updated the related works section to include the recommended methodologies, noting that most operate under fundamentally different settings compared to our approach. Below, we address the specific concerns and provide detailed comparisons:
>
> 1. **LLM-Based Test Generation and Fuzzing Techniques**
>
> **TitanFuzz, FuzzGPT, ChatUniTest, and Fuzz4All:**
>
> TitanFuzz and FuzzGPT are specialized deep learning libraries tailored for different testing requirements and do not align with our focus on Wasm binaries. ChatUniTest focuses on Java test case generation, but some of its features are incompatible with Wasm binaries. Lastly, while Fuzz4All is a generic framework, its application is hindered by the token length of Wasm binaries and LLMs’ limited ability to comprehend raw Wasm binaries effectively.
>
> 2. **Existing Wasm Testing Techniques**
>
> Several Wasm test generation approaches, such as **WasmMaker**, **Wasm-smith**, and **Wasm-mutate**, are designed for testing Wasm runtimes and compilers. These tools primarily generate syntactically correct Wasm code to fuzz Wasm-consuming programs. In contrast, our work focuses on testing Wasm code itself: WasmTest generates unit test cases for a given Wasm code to validate if it is performing the correct functionality and is bug-free.
>
> 3. **Detailed Comparisons:**
>
> To address the reviewer’s concerns, we conducted a comparative analysis of WasmTest and related methods. The results are summarized below, showcasing WasmTest’s performance against adaptations of existing techniques:
>
> - **Wasm decompilation works comparison:**
>
> We replace *Step 1: Generating Functionally Equivalent C++ Code* with Wasm decompilation models, which convert Wasm code to equivalent C/C++ code. Specifically, we compare WasmTest with the following binary decompilation tools one at a time. For each decompiler, we use the decompiled C/C++ code as an intermediate representation and prompt the LLM to **directly generate tests** based on the decompiled version.
>
> 1. WaDec [1] + direct generation of tests
> 2. StackSight [2] + direct
> 3. Wasm2c [3] + direct
>
> - **MuTAP integration:**
>
> We incorporate MuTAP [4] into our pipeline. Following the type-aware mutation step, we manually introduce errors into the functions under test to evaluate whether existing test cases can detect these injected bugs. For errors that remain undetected, we utilize LLMs by providing the function under test, the associated test cases, and details about the undetected errors. The LLMs are then prompted to generate new test cases specifically aimed at identifying the highlighted errors.
>
> Table 1: Metrics on HumanEval-X with one of the best performing models - CodeQwen1.5-7B.
>
> | Method              | C++ Compile | Wasm Compile | C++ Coverage | Wasm Coverage | Pass Rate |
> | ------------------- | ----------- | ------------ | ------------ | ------------- | --------- |
> | WasmTest            | **86.4%**   | **86.4%**    | **99.3%**    | **95.2%**     | **76.5%** |
> | WaDec + Direct      | 27.5%       | 28.4%        | 86.2%        | 93.3%         | 24.1%     |
> | StackSight + Direct | 54.81%      | 56.3%        | 99.1%        | 93.3%         | 22.4%     |
> | Wasm2c + Direct     | 0%          | 0%           | N/A          | N/A           | N/A       |
> | MuTAP               | 77.5%       | 77.1%        | 97.8%        | 92.3%         | 35.0%     |
>
> **Key Observations:**
>
> - Wasm2c: Fails to simplify complexities in Wasm binary (verbatim transcription), making it ineffective for guiding LLMs to generate compilable test cases.
> - WaDec and StackSight: Despite being state-of-the-art LLM-based Wasm decompilers, their lower compile and pass rates indicate challenges in achieving perfect decompilation.
> - MuTAP: Achieves the best compile and pass rates by leveraging LLMs to generate equivalent C++ code based on natural language summaries, which are easier for LLMs to process.
> - WasmTest: Outperforms all baselines across metrics due to its design, which incorporates sampling, majority voting, and type-aware mutations, ensuring superior compile rates, coverage, and pass rates.
>
> **Reference**
>
> [1] She, Xinyu, Yanjie Zhao, and Haoyu Wang. WaDec: Decompiling WebAssembly Using Large Language Model.
>
> [2] Fang, Weike, et al. StackSight: Unveiling WebAssembly through Large Language Models and Neurosymbolic Chain-of-Thought Decompilation.
>
> [3] Wasm2c. https://github.com/WebAssembly/wabt/wasm2c
>
> [4] Dakhel, Arghavan Moradi, et al. Effective test generation using pre-trained large language models and mutation testing.

---

> ### Author Response · Authors · 2024-11-30
>
> > **Q2:** **What are the applications of this black-box setting?**
>
> Generating tests for code snippets is crucial for ensuring software quality and reliability, yet it remains a tedious and time-consuming task for developers [1]. Similar to existing efforts in automated test generation [2, 3], our approach focuses on automatically producing a comprehensive test suite for a given program.
>
> WebAssembly (Wasm) is widely used across a wide range of applications. However, Wasm modules are often distributed as third-party binaries without accessible source code (a black-box setting) [4]. In some cases, these binaries may even be obfuscated to hide their actual functionality. According to [5], 28.8% of Wasm binaries are minified to conceal their true purpose. This poses a challenge for developers who need to integrate such Wasm modules into their applications, as they must ensure the module behaves as intended and is free of bugs. In such a setting, **developers** have no access to the source code and may require LLM-based tools for decompilation and test generation.
>
> In this black-box context, the input and output of the testing process can be described as follows:
>
> - **Input:** The program under test, accompanied by a description of its intended functionality provided in a natural language summary.
> - **Output:** A set of comprehensive unit tests, including test inputs and expected outputs, designed to validate the module’s correctness and expose potential bugs.
>
> This black-box approach is particularly valuable for developers working with precompiled or obfuscated Wasm binaries, enabling them to verify and trust the functionality of these components within their applications.
>
> **Reference**
>
> [1] Wang, Junjie, et al. Software testing with large language models: Survey, landscape, and vision. IEEE Transactions on Software Engineering, 2024.
>
> [2] Arghavan Moradi Dakhel, Amin Nikanjam, Vahid Majdinasab, Foutse Khomh, and Michel C Des- marais. Effective test generation using pre-trained large language models and mutation testing. Information and Software Technology, 171:107468, 2024.
>
> [3] Jiawei Liu, Chunqiu Steven Xia, Yuyao Wang, and Lingming Zhang. Is your code generated by chat- gpt really correct? rigorous evaluation of large language models for code generation. Advances in Neural Information Processing Systems, 36, 2024a.
>
> [4] Marius Musch, Christian Wressnegger, Martin Johns, and Konrad Rieck. New kid on the web: A study on the prevalence of webassembly in the wild. In Detection of Intrusions and Malware, and Vulnerability Assessment: 16th International Conference, DIMVA 2019, Gothenburg, Sweden, June 19–20, 2019, Proceedings 16, pp. 23–42. Springer, 2019a
>
> [5] Aaron Hilbig, Daniel Lehmann, and Michael Pradel. An empirical study of real-world webassembly binaries: Security, languages, use cases. In Proceedings of the web conference 2021, pp. 2696– 2708, 2021
>
> > **Q3:** **How does WasmTest verify the equivalence of LLM-generated C++ implementation? Can we apply reverse engineering tools to guide test case generation?**
>
> 1. **Equivalence Verification:**
>
> In the benchmark datasets used, we leverage the example tests provided in code summaries (e.g., Figure 2(a)) to filter out incorrect implementations. To further enhance robustness, we do not rely on one single implementation, instead, we adopt a *majority voting* approach across multiple implementations:
>
> - **Test Input Generation:** LLMs generate candidate test inputs.
> - **Expected Output Verification:** For each test input, if the majority of implementations produce the same output, this output is adopted as the expected result in the test case. Otherwise, the test case is discarded.
>
> 2. **Reverse Engineering for Wasm Binaries:**
>
> When a Wasm code summary is unavailable, reverse engineering can be applied to decompile the Wasm binary into a C++ implementation. However, this approach typically has lower accuracy due to the inherent complexities of Wasm binaries. As shown in Table 1, methods like StackSight, Wasm2c, and WaDec yield significantly lower compile rates and pass rates, reflecting the limitations of using reverse-engineered outputs in guiding test case generation effectively.

---

> ### Author Response · Authors · 2024-11-30
>
> > **Q4:** **Clarification regarding “Correct Test Output Prediction Rates”**
>
> We apologize for any confusion caused by our earlier explanation and thanks for pointing out the need for clarification. The 50% correctness rate reported in RQ5 (Table 7) refers to test outputs directly generated by LLMs, which often (~50%) include errors (e.g., isPrime(8) == True). Recognizing this limitation, our approach incorporates a **Test Case Correction** mechanism through majority voting, as detailed in Sections 3.2 and 3.3. This process significantly enhances the accuracy and reliability of the generated tests.
>
> **Test Case Correction:**
>
> 1. **Multiple Implementations:** We use multiple LLM-generated C++ implementations for the same functionality.
> 2. **Consensus-Based Output:** For each test input, the majority output from these implementations is adopted as the test output. If no majority consensus is reached, the test case is discarded.
>
> This correction mechanism mitigates the errors inherent in direct LLM-generated outputs, ensuring a higher degree of correctness and robustness. To prevent misunderstanding, we have updated our manuscript to clearly explain this mechanism and its role in overcoming the limitations of direct oracle generation by LLMs.
>
> > **Q5: Novelty in the mutation process**
>
> Our mutation process builds upon prior work [1] to systematically mutate test inputs and generate diverse test cases. However, unlike previous schemes designed for human-written tests, our mutation techniques are specifically optimized for various forms of LLM-generated tests. This includes unique structures such as function calls, container elements, and logical operations.
>
> While we acknowledge the potential for more innovative mutation techniques, our current approach demonstrates strong performance, achieving approximately 95% code coverage for both C++ and Wasm. Notably, the tests are able to detect carefully crafted bugs (negative examples) almost all the time, with DeepSeek-Coder-V2 showing exemplary performance with a bug detection rate of 98.7% on the MBXP dataset.
>
> We appreciate the reviewer’s feedback and will explore additional mutation strategies in the revised version to further enhance coverage and bug detection rates.
>
> [1] Jiawei Liu, Chunqiu Steven Xia, Yuyao Wang, and Lingming Zhang. Is your code generated by chat- gpt really correct? rigorous evaluation of large language models for code generation. Advances in Neural Information Processing Systems, 36, 2024a.

---

### Meta-Review · Area_Chair_5xub · 2024-12-22

**Metareview:**

> This paper introduces WasmTest, an approach for testing of WebAssembly (Wasm) binaries. To overcome the challenges of testing Wasm binaries without source code, WasmTest leverages a generate-then-test methodology. It utilizes large language models (LLMs) to automatically generate equivalent C++ code, create and mutate test cases, and compile these tests for execution against the Wasm binary. Results demonstrate that WasmTest generates comprehensive test suites.

The reviewers agree on experimental and applicability weaknesses. Even though the paper is interesting, its current scope is too limited for ICLR.

**Additional Comments On Reviewer Discussion:**

Only reviewers 52kk and xrYm engaged a bit in the rebuttal and did not change their score.

---

### Decision · Program_Chairs · 2025-01-22

Reject